# Vacuum Arc Plasma Coating for Polymer Surface Protection— A Plasma Enhanced In-Orbit Additive Manufacturing Concept

**Marina Kühn-Kauffeldt** [1,*,†] **, Marvin Kühn** [1,†] **, Michael Mallon** [2] **, Wolfgang Saur** [3] **and Fabian Fuchs** [4]

1 Institute for Plasma Technology and Mathematics, Universität der Bundeswehr München, Werner-Heisenberg-Weg 39, 85579 Neubiberg, Germany
2 Materials and Processes Section, Mechanical Department, ESA-ESTEC, Keplerlaan 1, NL-2201 AZ Noordwijk, The Netherlands
3 Institute of Construction Materials, Universität der Bundeswehr München, Werner-Heisenberg-Weg 39, 85577 Neubiberg, Germany
4 Wehrwissenschaftliches Institut für Werk- und Betriebsstoffe (WIWeB), Department 310-Surface Technology and Analytics, Institutsweg 1, 85435 Erding, Germany
* Correspondence: marina.kuehn-kauffeldt@unibw.de
† These authors contributed equally to this work.

**Abstract:** In-orbit additive manufacturing (AM) is a promising approach for fabrication of large structures. It allows to expand and accelerate human space exploration possibilities. Extrusion-based AM was demonstrated in zero gravity, while the realization of such a process in orbit-like vacuum conditions is currently under exploration. Still, a solution for protection of the UV and IR radiation sensitive polymers is needed in order to prevent their early mechanical failure under space conditions. Vacuum arc plasma based process is widely applied on earth for thin protective coating deposition. Its major advantage is its scalability—from tiny size used in electric propulsion to large scale coating devices. The usability of the vacuum arc process in space conditions was shown in electric propulsion applications in nano-satellites. In this work we discuss and demonstrate the integration of vacuum arc process as a post processing step after Fused Filament Fabrication (FFF) for additive manufacturing and functionalization of long polymer structures. Here we address the concept for technical realization, which integrates the vacuum arc into additive manufacturing process chain. More over we present a laboratory prototype, which implements this concept together with a use case, where a previously printed PEEK structure is coated with aluminum based coating suitable for UV radiation protection.

**Keywords:** vacuum arc; high temperature polymers; PEEK; in orbit manufacturing; vacuum; thin protective coating; FFF

## 1. Introduction

In the era of new space economy many efforts are put into the realization of the Earth orbit, Moon and Mars colonization. Here techniques for in orbit or on the Moon manufacturing, hence in the absence of the atmosphere are needed in order to be able to build large structures, which can not be transported by launchers from earth. Thus, the cost for in orbit and on the moon manufacturing can be reduced. In addition, utilization of available resources instead of resources sent from earth is considered in the further course. More over projects such as Solaris [1] aim to use space based technologies to achieve benefits for Earth. Here a large scale solar energy harvesting in orbit is intended in order to achieve higher rate of green energy on earth.

Filament based 3D printing is one of the technologies, which has proven to have many advantages for in-space manufacturing. It was already shown that it is suitable for zero gravity environment and it can produce structures which are larger than the manufacturing device itself. The filament in the 3D printer can be any kind of polymer.

Polyetheretherketon (PEEK) is an especially interesting polymer for space applications due to excellent mechanical and insulating properties, as well as it's resistivity to beta and gamma radiation. However PEEK and also other polymers have a poor ultraviolet (UV) radiation and atomic oxygen (AO) resistivity, which are considered one of the main reasons for the degradation of the surfaces exposed to the space environment [2,3]. This results in reduced mechanical properties, when experiencing exposure to UV light as well as material erosion due to AO. Also infrared (IR) radiation is more critical for polymer based structures than for those build from metal due to their lower melting temperatures and heat conduction properties. So care needs to be taken in order to protect polymer parts from UV and IR radiation.

Thin protective coatings composed from metal oxides and metals are valuable means of reducing oxidative attack as well as damage by the UV radiation [4]. Thin aluminum based coatings in the range of several 100 nm can provide such a protection [5]. Al is barely eroded by AO and has an excellent radiation reflectivity above 200 nm up to IR wavelength. For protection in lower wavelength region MgF or LiF protective layers can be added [6].

An other problem which space structures are facing is thermal control. Here as well multi-layer coating based solutions were developed [7] in order to passively compensate for thermal gradients in space structures. Such solutions need to be also integrated into an in orbit manufacturing process. Hence a process which is able to coat structures in orbit is needed in order to produce more radiation resistant polymer structures.

There is a variety of deposition methods operating on earth, capable of producing such coatings ranging from Sol-Gel, liquid phase deposition, cold and hot atmospheric plasma, chemical and physical vapor deposition techniques. However only the latter are applicable in low pressure conditions. Those techniques include different sputtering methods, laser ablation and vacuum arcs, from which only the latter two do not rely on presence of an additional gas. Yet, to date, non of the coating techniques used on earth were applied for on-orbit thin film deposition [8].

In the vacuum arc process, a solid metal cathode is evaporated and ionized. The so generated ions have a high velocity of approximately $1 \times 10^4 \, \mathrm{m\,s^{-1}}$ to $2.5 \times 10^4 \, \mathrm{m\,s^{-1}}$. When deposited on polymer substrate the ions are subplanted in the substrate surface e.g. the polymer matrix. Thus a good coating adhesion is achieved. One major advantage of the vacuum arc is, that it does not need additional gas to operate the plasma discharge. This is also the case for generation of coatings containing other components than metals. Metal oxide, nitrite etc. coatings can be deposited by incorporating these components in the cathode in the solid state, so that they can be evaporated during the plasma process operation [9]. Vacuum arcs can also be operated in pulsed mode. This on the one hand allows to coat temperature sensitive polymer substrates, while on the other hand the coating process can be operated with a low power budget, which make it potentially interesting for an on-orbit application [5].

Pulsed laser deposition uses focused laser beam to ablate and ionize the material and can produce similar quality coatings as vacuum arcs [10]. The major advantage of vacuum arc with respect to pulsed laser deposition is, that the deposition area can be easily enlarged by enlarging the cathode size. More over vacuum arcs can be easily scaled with the available power budget and they have already been operated in space environment as electric propulsion system for Cubsats [11].

In order to provide a possibility to efficiently build large and sustainable structures in space we suggest to operate polymer based additive manufacturing process in vacuum environment and combine it with an additional post processing coating step. Thus the coating could improve the polymer properties and extend the lifetime of polymer based structures in space. Here for we propose to use a small, lightweight and low power coating system based on a vacuum arc, which can coat outer surface of these structures. It is able to produce thin metal and metal oxide coatings from sintered cathode materials containing e.g. oxide components [12]. It can be integrated on the exit side of a 3D printer manufacturing these structures. As a protective coating a variety of material composition can be used. As

a first step we propose a use of an aluminum-based protective coating as the basic coating due to its excellent UV and AO resistant properties.

In this work we present a laboratory prototype of on orbit coating concept, where a miniature vacuum arc coating source integrated as a post processing step of a Fused Filament Fabrication (FFF) process. First the principle functionalities are discussed. Afterwards, the potential use case is presented. Here long structures with an aluminum coating for UV protection are manufactured.

## 2. Materials and Methods

### 2.1. FFF System

The implementation of the additive manufacturing process in vacuum environment requires a customisation of the FFF system. A prototype (Figure 1) with a build volume 35 mm × 35 mm × 65 mm (width × depth × height) was designed to match the vacuum chamber requirements. The kinematics were implemented as a belt driven rectilinear Cartesian system powered by Nema 17 motors. A full metal bowden extruder (Micro-Swiss, 24 V, 1.75 mm filament, 25 W heater cartridge) was used together with a custom made water-cooled hot-end assembly. A stainless steel nozzle (0.6 mm in diameter) was used in order to withstand high temperatures. All the mechanical components were chosen to be lubricant free to minimize outgassing in vacuum. All motors and the extruder were actively water-cooled (Innovatek, Stammham, Germany H2O complete module). The hardware was controlled by a Fystc Spyder mainboard. A BL-Touch (Ancatlabs) was used as a bed levelling sensor. The controller was operated with Klipper firmware. The mass of the FFF setup is around 1.5 kg. It however was not yet optimised concerning it's weight and space graded components.

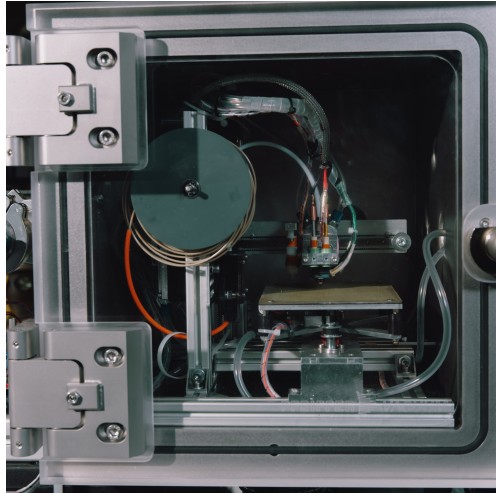

**Figure 1.** Photograph of the filament-based additive manufacturing system operating in vacuum.

### 2.2. Vacuum Arc Plasma Coating Unit

The vacuum arc plasma coating unit was designed to fit the space requirements of the FFF system. Figure 2 (left) shows the design of the coating apparatus, which allows the functionalization of the surface of the 3D-printed rod directly after the printing procedure. A single coating head is composed of a concentrically arranged cathode insulator and anode (Figure 2, (right)). Here, the coating head is fixed on the x-z-axis of the 3D printer kinematics, while the build plate is equipped with an additional rotary axis. Thus, the printed specimen can be rotated, ensuring that the entire surface is coated. Depending on the dimensions of the specimen cross section, the distance between the coating head and the specimen surface can be adjusted, so that the whole plasma plume hits the specimen surface. Thus material losses are minimized. The coating thickness is accurately controlled by the number of fired plasma pulses.

### 2.3. Power Supply Design

The power processing unit (PPU) needs to meet several requirements for this application. On the one hand, a lightweight and compact system working in the low power mode is needed, which would allow one to operate the coating unit on a small satellite such as a CubeSat composed of several sub-units. On the other hand, the PPU output has to ensure a reliable vacuum arc operation and maximise mass to input the charge rate. Hence, a capacitor-based storage system consisting of a three-staged capacitance pulse forming network was chosen, since it is able to fulfill all the upper requirements, as discussed in [13].

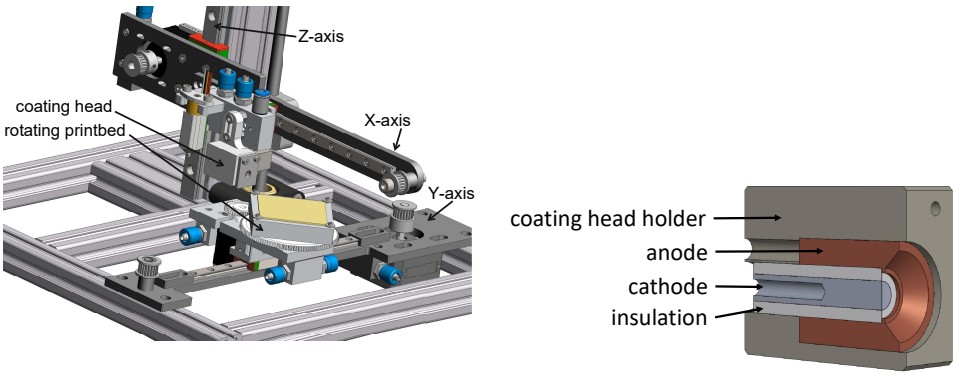

**Figure 2.** (**left**) Technical design of the post printing coating apparatus and (**right**) schematic setup of the vacuum arc coating source.

Figure 3 shows the schematic principle of the PPU and indicates the main functional blocks. The detailed electrical circuit is described in detail in [13]. It can be operated with the station power bus (24 V) and has a controllable pulse frequency output. A DC/DC converter is used to convert the bus voltage to the voltage required to initiate the discharge through the ignition unit in various trigger modes—high voltage surface discharge, and triggerless, fused ignition [9]. In the high voltage ignition mode, the resistance between the cathode and the anode is above 100 kΩ. In this case, the ignition unit is able to generate a voltage of up to 5 kV, which is sufficient to initiate the discharge. Once a conducting layer lowers the resistance to the kΩ range, the ignition unit operates in the so-called triggerless mode. Here, only voltage in the range of several hundred volts is needed to initiate the discharge through the conducting path. When the resistance between the cathode and the anode drops to a few Ω, the fused ignition mode is established. Here, in particular, the pre-arc discharge unit is required for a discharge stabilization directly after ignition. It can thus "defuse" the coating head, in case a considerable amount of material was deposited between the cathode and the anode leading to low resistance between the anode and the cathode or even to a short circuit. The main arc supply unit is responsible for providing the energy for the discharge, once a conducting plasma channel has been established. Although the PPU's frequency is fully controllable, it is limited by the available power budget. Pulse lengths can be adjusted by the choice of capacitance or other pulse forming elements. Active pulse shaping is possible. Table 1 summarizes the technical property range of this PPU design.

This kind of power supply was used for the operation of vacuum arc-based electric propulsion systems for Cubesats with a power budget in the range of 1 W. It was demonstrated that the vacuum arc system using this kind of PPU can be reliably operated over 10 million pulses [14]. In this configuration, the erosion rate of the 15 µg C$^{-1}$ was measured for a titanium cathode.

Since this technology is aimed to be applied on space missions, the SWAP-C (Space, Weight, Power and Cost) of the system needs to be minimized. For the proposed coating system, it results from the space and weight of the coating head itself, which can be as little as 100 g. In addition, the size of the PPU can be reduced to 60 mm × 60 mm × 25 mm with a weight ≤ of 100 g for a 1 W power budget. In the current setup, however, the PPU

dimensions are 100 mm × 100 mm × 35 mm with a weight of 122 g. Due to the simplicity of the coating head designed without moving parts, the manufacturing cost for the system are low (<1500 EURO including material cost and machining time). The cost for manufacturing the PPU are in the typical range for space-certified electronics. The costs scale linearly with the number of coating heads that are to be applied in the system. However, one PPU can be used for several coating heads simultaneously, which can be either used to accelerate the deposition rate or as a replacement, once one coating head reached the end of its life cycle.

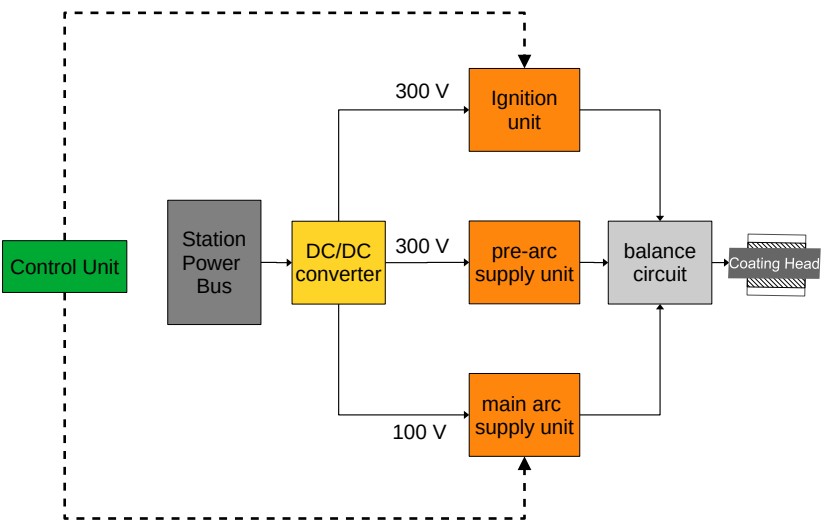

**Figure 3.** Schematic of the PPU.

**Table 1.** Overview of possible operation parameter of the PPU.

| Properties | Value (Range) |
| :---: | :---: |
| Power consumption/budget | 0.5 W to 30 W |
| Pulse lengths | 50 µs to 5000 µs |
| Repetition rate | 0 Hz to 10 Hz |
| Output current | 20 A to 300 A |
| Output voltage | 150 V to 5000 V |
| Input current | 0.1 A to 3 A |
| Input voltage | 24 V |
| Minimal size (length × width × height) | 60 mm × 60 mm × 25 mm |
| Minimal mass | 100 g |

*2.4. Vacuum Setup*

In order to test the suggested prototype in laboratory conditions, the FFF-system equipped with the post-processing coating unit was positioned inside a stainless steel vacuum chamber (inner dimensions 300 mm × 300 mm × 300 mm). For vacuum generation, a HiPace 80 Turbo Pump together with a Duo 5 m backing pump were used (Pfeiffer Vacuum, Assler, Germnay). The pressure was monitored via a PKR 251 wide range pressure sensor (Pfeiffer Vacuum, Assler, Germnay). The FFF hardware was electrically connected via standard electrical feedthroughs to the electronic boards that were located outside the vacuum chamber. For the water-cooling circuit, flexible rubber tubes were used to link the individual serially arranged heatsinks to a radiator. In this way, sufficiently outgassing the proof flexible connections between the heatsinks and feedthroughs could be ensured. The vacuum setup containing the FFF-system could reach a pressure in the range of $1 \times 10^{-5}$ mbar without the operation of the extruder unit.

*2.5. Use Case*

For the first demonstration of the system, a long tube structure was chosen. It was first manufactured in vacuum conditions and was then coated with an aluminum-based

coating. Here, a simple single walled PEEK tube was chosen as a printing object, as shown in Figure 4. It was printed in the so-called spiral vase pattern, where the printer continuously extrudes the filament. Thus, a specimen without seams is obtained. The tube diameter and height were chosen with 12 mm and 30 mm, respectively. The filament PEEK (INFINAM® PEEK 9359 F, Evonik, Essen, Germany) was used as received. Printing and coating parameters for the specimen are summarised in Table 2. The specimen was printed at the pressure of $4 \times 10^{-4}$ mbar. The nozzle temperature was 365 °C, while no additional heating was applied (bed, chamber).

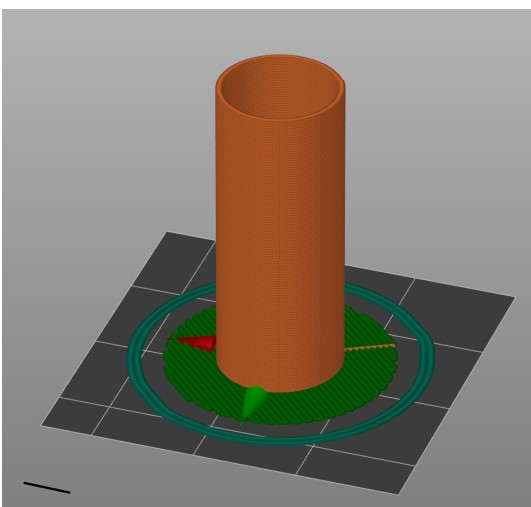

**Figure 4.** Printing model of the tube printed as a single wall object with the spiral vase method (scale bar 5 mm).

**Table 2.** Printing and coating parameters of PEEK specimen.

| Parameter | Value | |
|---|---|---|
| | *printed substrate* | *wafer* |
| Operating pressure [mbar] | $4 \times 10^{-4}$ | $4 \times 10^{-4}$ |
| Material | PEEK | Si |
| Nozzle temperature [°C] | 365 | - |
| Chamber temperature [°C] | room temperature | room temperature |
| Nozzle speed [mm s$^{-1}$] | 5 | - |
| Layer height [mm] | 0.2 | - |
| Extrusion width [mm] | 0.65 | - |
| Manufacturing time [min] | 17 | - |
| | *coating* | |
| Coating material | Al | Ti |
| Cathode diameter [mm] | 3 | 3 |
| PPU main capacitor [µF] | 144 | 144 |
| Main charging voltage [V] | 100 | 100 |
| Operation frequency [Hz] | 2 | 2 |
| Rotation step [°/pulse] | 0.25 | - |
| Total number of pulses | 14,400 | 1000 to 2000 |
| Cathode-specimen distance [mm] | 8 | 30 |

The coating procedure was performed at the same pressure range with an aluminum cathode at a coating head to a specimen distance of 8 mm. The coating head was not moved in the z direction, so that the difference between the coated and uncoated region could be visualized. The specimen was rotated 0.25 °/pulse in order to ensure a homogeneous coating thickness. The PPU was operated with a main capacitor of 144 µF, which was charged

to 100 V in order to achieve sufficient erosion rate at a power budget approximately 1 W. In order to deposit a clearly visible coating, a total number of 14,400 pulses was required.

In addition to the use case, the deposition on silicon wafers was performed in order to characterise the possible deposition rates achievable with such a PPU and to investigate the morphology and distribution of the coating layer on an ideal surface. Here, the set up was operated with the same parameters, replacing the cathode by titanium in order to be able to perform coating characterisation based on the weight difference between the bulk and coating atoms. Wafers with an area of 15 mm × 15 mm were coated. For the analysis, the coated silicon wafers were fractured along their crystal orientation. Hence, a relatively sharp waste edge was obtained.

### 2.6. Coating Analysis

The coated specimen surface was analyzed by means of a scanning electron microscope (SEM) model Zeiss EVO LS15 in the high vacuum mode at an acceleration voltage of 20 kV. Before SEM observations, all sample surfaces were sputtered with an approx. 10 nm carbon layer.

The composition of the specimen surface was analyzed by means of an Energy Dispersive X-Ray (EDX) analysis using the Oxford Instruments X-MaxN 50 EDX-System with a SDD-detector (Si(Li)-Detector) with 50 mm$^2$. Here, the acceleration voltage was 20 kV.

Moreover, a field emission SEM model Zeiss ultra plus with an InLense detector was used to investigate the coating wafer interface. Here, no additional coating was applied and the acceleration voltage of 1 kV was chosen.

In addition, the surface structure of the printed specimen was analysed using a laser scanning microscope (LSM, VK-X 3000, Keyence, Neu-Isenburg, Germany).

## 3. Results and Discussions

### 3.1. Additive Manufacturing and Coating of Polymer Structures

The FFF-setup was successfully operated in a vacuum and thus demonstrated the 3D printing of the PEEK structure together with the operation of the coating process. As mentioned in Table 2, the working pressure was p = $4 \times 10^{-4}$ mbar. In this pressure range, the mean free path is in the range of 50 cm [15], which still allows one to assume the coating process as collisionless. However, since according to the ideal gas law ($p = nk_BT$ with $k_B$, the Boltzman constant and $T = 20\,°C$), the particle density is still in the range of n = $1 \times 10^{19}$ m$^{-3}$, and the aluminum will most likely oxidize, building a mixture of aluminum and aluminum oxide coating. In Figure 5 (right), the resulting coated specimen is shown. From the uncoated regions on the top and the bottom of the specimen, it can be deduced that the printed PEEK structure is homogeneously crystalline. The coating can be clearly distinguished as a gray area indicated in the image.

Figure 5 (left) shows the vacuum arc coating process being applied after the 3D printing process has been completed. Here, the blue plasma plume is clearly visible. The plume has a diverging shape with slightly bigger dimensions than the specimen diameter, which leads to material losses and thus to a less efficient deposition rate. In previous investigations, it was demonstrated that it is possible to focus the plasma plume by applying an external magnetic field [16]. This would allow one to minimize material losses. In order to implement the focusing, typically a solenoid positioned concentrically with respect to the cathode is used. It is operated in series with the arc discharge and increases the overall mass by only 2 g to 3 g. In order to have the best operational mode, the optimal inductance as well the relative position needs to be experimentally determined.

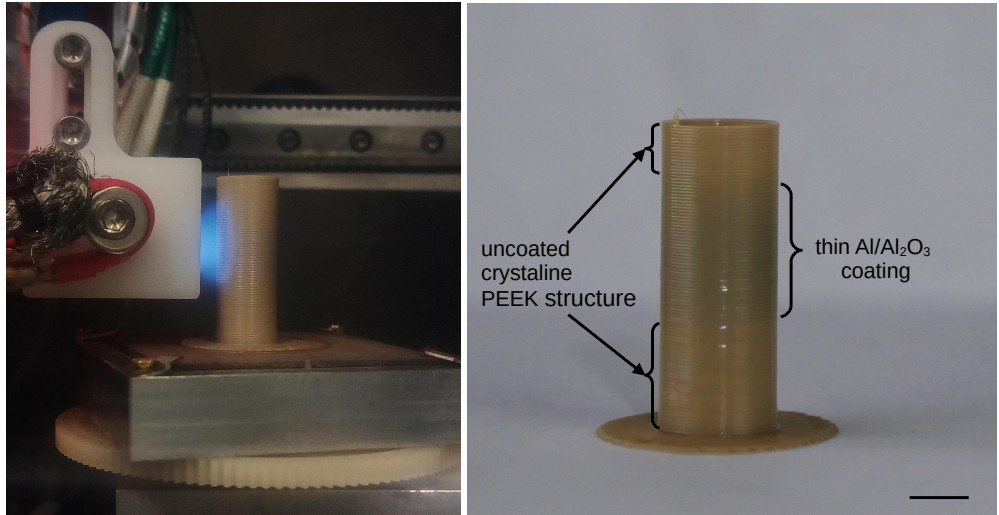

**Figure 5.** (**left**) Photograph of functionalization of the 3d printed structure by the vacuum arc plasma coating process; (**right**) PEEK tube printed and coated at the pressure of $1 \times 10^{-4}$ mbar (scale bar 5 mm).

### 3.2. Coating Characterization

Figure 6 shows the height profile of the 3D-printed specimen structure, on which the protective coating was deposited. Here, the wave-like surface has a height difference in the range of 60 μm, while the wave peaks separation is determined by the printing layer height and is in the range of 250 μm. Moreover, a surface roughened below 1 μm was determined from the measured profile. The inclination angle of the filament strand sides is approximately 35°.

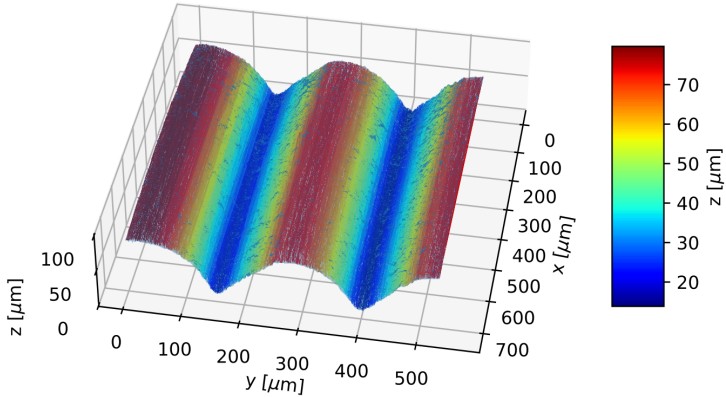

**Figure 6.** Surface profile of the 3D-printed specimen recorded with an LSM.

Generally speaking, the surface of the specimen is sufficiently flat for a good coating adhesion. The resulting coating is visualized by means of an SEM and EDX analysis, as shown in Figure 7. Here, macroparticles typical for unfiltered vacuum arcs can be observed on the left image over the whole slope of the filament strand. On the right, two sides of the SEM image are overlaid with the spatially resolved EDX signal of the detected elements C, O and Al. Here, a homogeneous distribution of all the elements can be observed on the slope of the deposited filament strand on the right of the image. The signal for all the elements is absent in the center of the image. This is due to the fact that the EDX detector is positioned at an angle of 40° and thus the area in the center of the image is shadowed by the next filament strand.

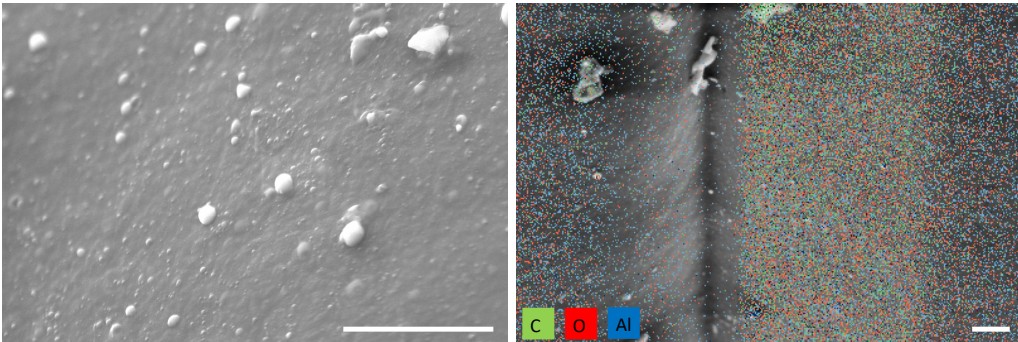

**Figure 7.** (**left**) SEM micrographs of the Al coating deposited on the printed specimen (**right**) together with an overlay of the spatially resolved EDX signal detected on the specimen surface (scale bar 20 µm).

The relatively transparent color of the coating in Figure 5 indicates that it mainly contains $Al_2O_3$. The coating composition could be confirmed from EDX spectra comparison taken at different parts of the specimen (Figure 8). In the central area, the Al and O peak clearly dominate over the C peak coming from the bulk material, while at the edge mainly C and O are detected being the main detectable components of PEEK (chemical formula $C_{19}H_{12}O_3$). Although it is not possible to quantify the stoichiometry of the coating from the spectra due to the presence of Oxygen in the coating and in the bulk, the considerably higher peak in the center of the specimen suggests that Al is oxidized. The light Al signal in the edge spectrum can be explained by small amount of plasma and macroparticles, which are scattered during the deposition process.

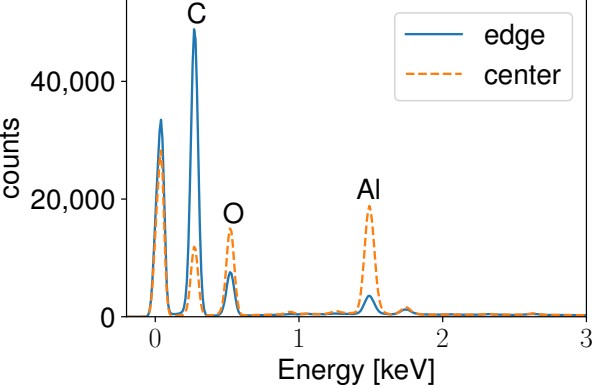

**Figure 8.** EDX Spectrum taken in the main coating area at the center and in the uncoated region at the edge of the specimen.

Figure 9 shows the cross-section of the silicon wafer coated with Ttanium using the same PPU setup. Here, the samples were coated with three different pulse numbers. The field emission SEM micro-graphs demonstrate a homogeneous coating, which generates a stronger signal than the bulk material (Si) due to considerably higher atomic mass of the coating (Ti). A homogeneously distributed coating layer can be visualized in all the three samples. In the sample B, a macroparticle is situated on top of the coating. From this series, an average deposition rate of the 0.057 nm/pulse was determined.

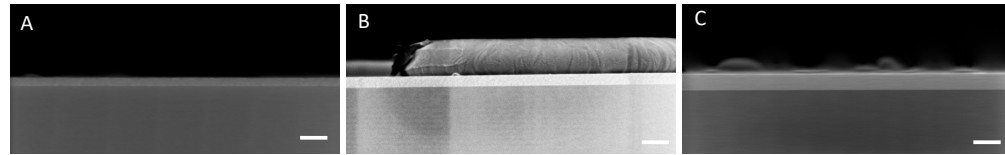

**Figure 9.** Cross-section of titanium coating deposited on a silicone wafer deposited with 900 (**A**), 1400 (**B**) and 1900 (**C**) pulses acquired using FEM. Scale bar corresponds to 200 nm.

*3.3. PPU Performance*

During the coating process, the performance of the PPU was evaluated in order to estimate the time per area necessary to deposit 100 nm of coating, which is a typical order of magnitude for the coating thickness needed for the radiation protection. Figure 10 (left) shows the typical current and voltage pulses generated during the coating process. Here, the peak current of around 170 A is reached, while the peak ignition voltage of approximately 270 V is needed to ignite the plasma. For a pulse length of 100 μs, this results in an average current of 45 A. From the voltage trace measured at the main arc supply unit voltage (Figure 10 (right)), the input power can be determined. Here, the energy is mainly stored in the main arc supply unit ($C = 144$ μF). The voltage at the main unit drops from $U_{init} = 100$ V to an arc burning voltage of around $U_{end} = 35$ V. Hence, the energy consumed during the single pulse equals to

$$E_{PPU} = \frac{1}{2}C(U_{init}^2 - U_{end}^2) \tag{1}$$

For the given values $E_{PPU} = 0.63$ J is obtained. At the operation frequency of 2 Hz, it corresponds to an average power of 1.26 W, which corresponds to the upper requirement. The deposition rate can be estimated here from the following equation

$$\dot{M}_{depos} = \frac{M_{cathode}}{t} = I_{arc}\gamma_c d_{pulse} \tag{2}$$

with $I_{arc}$ the average arc current, $\gamma_c$ the erosion rate of the cathode and $d_{pulse}$ the duty cycle of the pulse. For the titanium, we have previously determined an erosion rate of $\gamma_c = 15$ μg C$^{-1}$ with a comparable PPU setup [13]. With this data, a deposition rate of 0.135 μg s$^{-1}$ is obtained. The erosion rate value determined for our PPU is below the values given in the literature [17]. Hence, we also expect the erosion rates for aluminium to be lower than the literature values.

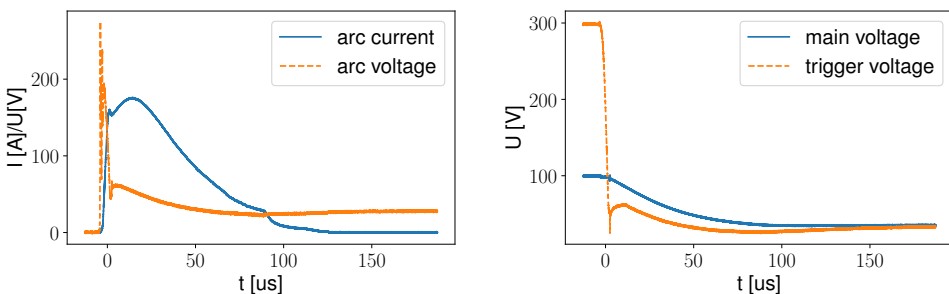

**Figure 10.** Current and voltage pulses measured (**left**) at the PPU output and (**right**) at the main and trigger units during the coating processes.

In order to estimate the time necessary to coat the entire surface of the specimen with a surface area of around A = 1000 mm$^2$ with s = 100 nm coating, the deposited mass of the coating can be calculated by

$$M_{depos} = \rho_c As \tag{3}$$

Here, we use the data that we have determined for a titanium cathode. Although for different materials such as aluminum, the precise values would be different, the order of magnitude of the results will remain the same. Here, $\rho_c = 4500\,\mu g/mm^3$ is the density of bulk titanium and $s$ is the thickness of the coating. Hence, the combination of the Equations (2) and (3) results in a coating time of approximately 55 min. This estimate does not takes into account the material losses, as was discussed in the previous section.

When using deposition rates determined from the deposition of titanium on silicon wafers and assuming that the area of $15 \times 15\,mm^2$ was homogeneously coated, it would consequently take around 65 min to coat an area of $1000\,mm^2$ with a 100 nm thick coating at the determined deposition rate of 0.057 nm/pulse and at the coating frequency of 2 Hz.

Both estimates lie in the same order of magnitude as the time needed for the manufacturing of the specimen itself, as indicated in Table 2. Hence, if the specimen is manufactured as an endless structure, the two processes could be operated simultaneously. This means that the already manufactured part of the endless structure can be coated, while the 3D printer continues the manufacturing.

It should be noted that the deposition rates scales with a current of the arc [18] hence with the applied power. If the power is up-scaled, e.g., to a typical power of a sputtering or a pulsed laser deposition system comparable, at least comparable or even superior deposition rates can be easily achieved [19]. Hence, this concept can be used as a low power coating source and is still competing with other deposition methods when it comes to deposition rates in the high power mode.

## 4. Conclusions and Outlook

In this work, it was demonstrated that the filament-based additive manufacturing can be combined with low power vacuum arc coating processes. It was further demonstrated that the deposition rates reachable at this power range are matching well the manufacturing speed used for the polymer structure fabrication under vacuum condition.

In the next step, the coating source will be further characterized in order to determine a parameter range suitable for in orbit applications. Moreover, we would like to determine optimal coating parameters, under which UV protective coatings suitable for in orbit conditions can be deposited. Here, properties of the deposited coating need to be further investigated with respect to their UV protection of polymers such as PEEK in orbit conditions (thermal cycling).

**Author Contributions:** Conceptualization, M.K., M.K.-K. and M.M.; investigation and validation, M.K., W.S. and F.F.; writing—original draft preparation; writing—review and editing, M.K. and M.M.; formal analysis, visualization, project administration and funding acquisition, M.K-K. All authors have read and agreed to the published version of the manuscript.

**Funding:** This research was funded by European Space Agency (Discovery program, ESA Contract No. 4000137515/22/NL/GLC/ov).

**Institutional Review Board Statement:** Not applicable.

**Informed Consent Statement:** Not applicable.

**Data Availability Statement:** Not applicable.

**Acknowledgments:** The authors acknowledge the Laboratory for High Power Electronic Systems at the Universität der Bundeswehr Munich for providing the laboratory infrastructure. The authors also thank the Institute of Construction Materials at the Universität der Bundeswehr Munich and the Department 310 Surface Technologies and Analytics at Wehrwissenschaftliches Institut für Werk- und Betriebsstoffe (WIWeB) for providing access to their imaging facilities.

**Conflicts of Interest:** The authors declare no conflict of interest. The funder had no role in the design of the study; in the collection, analyses, or interpretation of data; in the writing of the manuscript; or in the decision to publish the results.

## Abbreviations

The following abbreviations are used in this manuscript:

| | |
|---|---|
| FFF | Fused Filament Fabrication |
| PPU | Power Processing Unit |
| AO | Atomic Oxygen |
| SWAP-C | Space, Weight, Power and Cost |
| LEO | Low Earth Orbit |
| PEEK | Polyetheretherketon |
| UV | Ultraviolet |
| IR | Infrared |
| SEM | Scanning Electron Microscopy |
| EDX | Energy Dispersive X-ray spectroscopy |
| LSM | Laser Scanning Microscope |

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
