# Peer review of "Vacuum Arc Plasma Coating for Polymer Surface Protection— A Plasma Enhanced In-Orbit Additive Manufacturing Concept"

_plasma, doi:10.3390/plasma5040035_

Round 1

Reviewer 1 Report

This paper describes a prototype system that integrates filament-based 3D printing with plasma arc coating. The prototype is designed to operate in a vacuum environment, demonstrating the feasibility of in-orbit manufacturing. In this study, the experiments were carefully conducted, and the experimental setup is described in detail. The manuscript is in general well written. The new design can be of interest to the plasma community. I recommend its publication in plasma after a minor revision with the following comments addressed.

1.     In the introduction, I suggest the authors include an overview of current in-orbit coating systems and highlight the novelty and advantage of the authors’ new design over current systems.  

2.     Section 3.1: The authors claim that the coating was successful based on the visual color change in Figure 5. The color change is not strong evidence that aluminum oxide is deposited. For example, polymer degradation under the plasma could also contribute to color change. As the authors mention in Line 198, Energy Dispersive Xray spectroscopy can be used to determine whether aluminum or aluminum oxide has been deposited.

3.     Section 3.2: The authors evaluate the system performance based on aluminum corrosion rate and deposition rate from the literature (Page 8, Line 213). It would be more accurate to quantify the deposition rate based on the actual deposition rate from the prototype. For example, the authors can use the weight difference post/pre treatment to determine the deposition rate.

Author Response

Thank you very much for the valuable suggestions for improvement of our manuscript. Here are the answers to your comments.

  1. There is now current in-orbit coating system being used to our knowledge. We have included a comment on this in line 51, as well as a reference to a study discussing manufacturing technologies currently being considered for in orbit manufacturing.
  1. EDX analysis was added to the paper
  1. In order to conduct estimations with data close to our experiment, we now perform the estimation for titanium cathode, for which we have determined the erosion rates in previous work for comparable setup. More over we have determined per pulse deposition rate for Titanium on a silicon wafer, which we also present in the paper to provide a comparison.

Reviewer 2 Report

1. Use either kelvins or degrees Celsius as temperature units in the text.

2. In equation (1) for Uinitial and Section 3.2 for voltage at the main unit drops from Uinit use the same abbreviation.

3. Make the plots in Figure 6. readable for black-and-white printing.

4. Fig. 6. In the figure caption remove the bracket.

5. Write in equation (3) that the character A is used to designate the area.

6. How does the waveform relief of PEEK that is caused by the layer-by-layer patterning of the polymer during 3D printing influence on the adhesion of a 100 nm thickness aluminium coating?

7. Line 34. Instead of "Atomic Oxygen (AO)" it should be "atomic oxygen (AO)"

8. Line 19. Instead of "of earth orbit" it should be "of the Earth orbit".

9. Line 20. Instead of "moon and mars"  it should be "Moon and Mars".

10. Line 126. Instead of "triggerles" it should be "triggerless".

11. Line 185. Instead of "Atructures" it should be "Structures".

12. Line 198 Instead of AL2O3  it should be Al2O3.

13. Line 199. Instead of "Xray" it should be "X-ray".

14. Figure 3 should be located after the description in the text of the article.

15. In table 1 give the correct size for minimal size.

16. Figure 4 is sketched in isometry. Therefore the direction of the scale bar should be indicated accordingly.

17. How was the uniform thickness of the coating achieved? And how was the thickness of the coating measured on the wavy surface of the substrate?

18. How much will the use of magnetic lenses to focus the plasma increase the mass of the design? Is it reasonable to use magnetic fields to focus plasma in orbit to minimize loss of material?

19.  Line 214. Specify the combination of which two equations allows the estimation of time.

20. Indicate the total mass of equipment, i.e. 3D printer and Vacuum Setup, that was used to build the prototype. 

Author Response

Thank you very much for your comments which help us improve the quality of our manuscript. Please find detailed answers to your comments below.

Comments 1-5 were corrected as requested

  1. We have added profile measurements of the specimen waveform and have discussed the coating distribution on it in section 3.2. We consider that the coating thickness might vary slightly over the wave profile. However, the coating does not necessarily havr to be homogeneous but sufficiently thick to have UV protecting properties.

Comments 7-16 were corrected as requested.

  1. For Aluminum coating we do not claim that the coating thickness is uniform. We only demonstrate in section 3.2 that all of the surface was coated with aluminum. In order to measure the coating thickness either white light interferometry or coating cross-section generated by ion milling need to be used.
  1. We have added the estimated mass of the solenoid in the last paragraph of the section 3.1. Given the fact that the expected mass increase is only 2-3g we consider it reasonable to use magnetic fields for improvement of plasma trajectories.
  1. Specification was added in the line.
  1. We have indicated the weight of the 3D Printer in section 2.1. The Vacuum Setup weights around 80 kg. However we do not consider this information relevant to the replicability of the experiment, since any kind of laboratory set-up can be used to generate vacuum for on earth demonstration. The vacuum setup will not be needed on orbit.

Reviewer 3 Report

The paper centers around the the integration of vacuum arc process as a post-processing step after Fused Filament Fabrication (FFF) for additive manufacturing and functionalization of long polymer structures. Although it is potentially interesting for readers, it needs to include more experiments to make this manuscript meaningful and complete enough for readers.  I would suggest containing more characterizations such as SEM for morphological investigation, UV degradation, chemical analysis, and adhesivity. WIth this, I would like to reject this manuscript.

Author Response

Thank you very much for providing the suggestions to include coating characterization in the paper, which helped us to improve the quality of our manuscript. We have included SEM and LSM analysis for morphological of the coating and EDX for chemical analysis. This in our opinion demonstrates the general applicability of the technique for in orbit manufacturing of thin films. At this stage, we do not focus on the coating functionality such adhesivity and UV degradation yet. This is an application we would like to focus in the next step, where we would like to develop a UV protective coating for PEEK structures by means of low power vacuum arcs and investigate coating performance as well as its stability in space conditions.